# Genetic Background Underlying 5-HT_1A_ Receptor Functioning Affects the Response to Fluoxetine

**DOI:** 10.3390/ijms21228784

**Published:** 2020-11-20

**Authors:** Elena M. Kondaurova, Alexander Ya. Rodnyy, Tatiana V. Ilchibaeva, Anton S. Tsybko, Dmitry V. Eremin, Yegor V. Antonov, Nina K. Popova, Vladimir S. Naumenko

**Affiliations:** Department of Behavioral Neurogenomics, Institute of Cytology and Genetics, Siberian Division, Russian Academy of Sciences, Novosibirsk 630090, Russia; arodnyi@bionet.nsc.ru (A.Y.R.); ilchibaeva@bionet.nsc.ru (T.V.I.); antontsybko@bionet.nsc.ru (A.S.T.); dima.969696@mail.ru (D.V.E.); yegor@bionet.nsc.ru (Y.V.A.); npopova@bionet.nsc.ru (N.K.P.); naumenko2002@bionet.nsc.ru (V.S.N.)

**Keywords:** SSRI, fluoxetine, antidepressants resistance, 5-HT_1A_ receptor, mice, serotonin, Freud-1

## Abstract

The influence of genetic background on sensitivity to drugs represents a topical problem of personalized medicine. Here, we investigated the effect of chronic (20 mg/kg, 14 days, i.p.) antidepressant fluoxetine treatment on recombinant B6-M76C mice, differed from control B6-M76B mice by CBA-derived 102.73–110.56 Mbp fragment of chromosome 13 and characterized by altered sensitivity of 5-HT_1A_ receptors to chronic 8-OH-DPAT administration and higher 5-HT_1A_ receptor mRNA levels in the frontal cortex and hippocampus. Significant changes in the effects of fluoxetine treatment on behavior and brain 5-HT system in recombinant B6-M76C mice were revealed. In contrast to B6-M76B mice, in B6-M76C mice, fluoxetine produced pro-depressive effects, assessed in a forced swim test. Fluoxetine decreased 5-HT_1A_ receptor mRNA levels in the cortex and hippocampus, reduced 5-HT_1A_ receptor protein levels and increased receptor silencer Freud-1 protein levels in the hippocampus of B6-M76C mice. Fluoxetine increased mRNA levels of the gene encoding key enzyme for 5-HT synthesis in the brain, tryptophan hydroxylase-2, but decreased tryptophan hydroxylase-2 protein levels in the midbrain of B6-M76B mice. These changes were accompanied by increased expression of the 5-HT transporter gene. Fluoxetine reduced 5-HT and 5-HIAA levels in cortex, hippocampus and midbrain of B6-M76B and in cortex and midbrain of B6-M76C; mice. These data demonstrate that changes in genetic background may have a dramatic effect on sensitivity to classic antidepressants from the Selective Serotonin Reuptake Inhibitors family. Additionally, the results provide new evidence confirming our idea on the disrupted functioning of 5-HT_1A_ autoreceptors in the brains of B6-M76C mice, suggesting these mice as a model of antidepressant resistance.

## 1. Introduction

Depression is the one of the most common human mental disorders, significantly reducing the quality of life. Depression ranks second after cardiovascular diseases in terms of the number of days of a patient’s disability. However, despite the huge number of research studies being performed, the mechanisms and pathogenesis of depressive disorders are still far from having a complete understanding.

Many hypotheses on the mechanisms of depression indicate the key role of the brain serotonin (5-HT) system [1,2,3]. Classical clinically used antidepressants from the Selective Serotonin Reuptake Inhibitors (SSRIs) group are aimed at blockade of the serotonin transporter (5-HTT), which performs reuptake of the neurotransmitter from the synaptic cleft back into the presynaptic terminal. However, a significant percentage of depressive patients are resistant to this class of drugs, suggesting the role of individual genetic features. In addition, for the effective correction of depressive behavior, classical antidepressants must be utilized chronically. A number of studies indicate the implication of the serotonin 5-HT_1A_ receptor in the positive effect of chronically administered SSRIs [4]. These 5-HT_1A_ receptors could be localized both pre- and postsynaptically. Receptor function is strongly dependent on localization: in the raphe nuclei of the midbrain, it acts as a presynaptic somatodendritic autoreceptor, inhibiting neuronal activity and 5-HT secretion into the synaptic cleft, whereas the postsynaptic 5-HT_1A_ receptor mediates 5-HT action on neurons. There are many data demonstrating the involvement of 5-HT1A receptors in the mechanisms of depression [5,6,7,8,9,10], depressive psychosis [11] and suicidal behavior [12,13,14]. There is some evidence indicating the role of presynaptic 5-HT_1A_ receptors in the mechanism of antidepressant drug action [10,15,16]. An increased density of presynaptic 5-HT_1A_ receptors was observed in depressed patients and suicidal victims [14,17]. The combination of 5-HT_1A_ autoreceptor antagonists with classical antidepressants has been proposed to be more effective for depression treatment [18]. Preclinical data also show that activation of postsynaptic 5-HT_1A_ receptors is important for the antidepressant effect of 5-HT_1A_ receptor agonists [19]. Selective overexpression of postsynaptic 5-HT_1A_ receptors is associated with a clear-cut antidepressant response to the SSRI citalopram [20].

In 2000, in the promoter of the 5-HT_1A_ receptor gene, a binding site (dual repressor element-(DRE-element)) for a selective repressor suppressing the receptor gene expression was revealed [21]. Later, the transcriptional factor *Cc2d1a*/Freud-1 which binds to this DRE-element was identified [22]. It was shown that the removal of the promoter fragment containing the DRE-element results in a considerable increase in 5-HT_1A_ receptor gene expression. On the contrary, overexpression of *Cc2d1a*/Freud-1 suppresses 5-HT_1A_ receptor gene expression and reduces the level of the 5-HT_1A_ receptor protein. *Cc2d1a*/Freud-1 mRNA was found in the midbrain raphe nuclei, hippocampus, frontal cortex and hypothalamus. Freud-1 protein was shown to be co-localized with the 5-HT_1A_ receptor [22]. There are several studies suggesting Freud-1 involvement in the development of psychopathologies [23,24]. On the other hand, the functional state of 5-HT_1A_ receptors can be modified by other serotonin receptors. For example, 5-HT_2A_ receptors can form heterodimers with a 5-HT_1A_ receptor [25], affecting its functioning [26,27]. The 5-HT_7_ receptor is also able to form heterodimers with 5-HT_1A_ receptors, significantly changing the function of the latter [16,28].

Accumulating evidence suggests that different genetic backgrounds can affect the function of the same genes differently, resulting in alterations in responses to drug treatment [29]. The investigation of gene networks’ influence on response to drug treatment is an important problem of modern neuroscience. Mice created by a small genome fragment transfer represent promising models for studying the effects of genetic modification on molecular mechanisms of behavior and responses to drug treatment. Recently, we created recombinant B6.CBA-D13Mit76C (B6-M76C) and B6.CBA-D13Mit76B (B6-M76B) mouse lines on the C57Bl/6 genetic background [30,31]. The B6-M76C mice were differed by 5-HT_1A_ receptor sensitivity to chronic activation with 5-HT_1A_ receptor agonist 8-OH-DPAT, which allowed us to assume that B6-M76C mice have a genetically defined reduced sensitivity of presynaptic 5-HT_1A_ receptor [31] and that these changes in genetic background can modulate the response to 5-HT_1A_-related antidepressants from the SSRIs family. 

Here, to further confirm this idea, we studied the effects of chronic treatment with the classical SSRI antidepressant fluoxetine on (i) behavior in the open field and forced swim tests; (ii) mRNA and protein levels for Freud-1, 5-HT_1A_, 5-HT_2A_ and 5-HT_7_ receptors as well as TPH-2, MAOA and 5-HTT and (iii) 5-HT and 5-HIAA levels and 5-HIAA/5-HT ratio in the brain structures (frontal cortex, hippocampus, hypothalamus and midbrain) of B6-M76B and B6-M76C mouse lines.

## 2. Results

### 2.1. Open Field Test (OF)

Chronic fluoxetine treatment increased the path length in both B6-M76C and B6-M76B mice compared with correspondent control groups (F_1.58_ = 22.34, *p* < 0.001 for effect of fluoxetine; F_1.58_ = 6.71, *p* < 0.05 for effect of genotype) (Figure 1A). For time spent in the center of the arena, two-way analysis revealed an effect of fluoxetine (F_1.58_ = 7.04, *p* < 0.05), but post hoc analysis showed only a tendency for both B6-M76C and B6-M76B mice compared with the corresponding control groups (Figure 1B). 

### 2.2. Forced Swim Test

Chronic fluoxetine treatment led to decrease in mobility only in the B6-M76C mice (F_1.49_ = 5.71, *p* < 0.05 for effect of genotype x fluoxetine interaction; for effect of fluoxetine F_1.49_ = 3.59, *p* = 0.06 was at tendency level; however, post hoc analysis revealed significant differences in mobility, *p* < 0.05) (Figure 2). The B6-M76C mice demonstrated higher mobility compared with the B6-M76B mice (post hoc analysis *p* < 0.05) (Figure 2). Thus, despite an increase in locomotor activity, chronic treatment with fluoxetine resulted in enhancement of depressive-like behavior in B6-M76C mice, reflected in reduced mobility in the forced swim test.

### 2.3. Expression of Genes

The B6-M76C mice demonstrated higher levels of *Htr1a* gene mRNA in the frontal cortex (*p* < 0.05) and hippocampus (*p* < 0.05) compared with the B6-M76B mice. Fluoxetine decreased *Htr1a* gene mRNA levels in B6-M76C mouse frontal cortex (F_1.25_ = 12.64, *p* < 0.001 for effect of genotype; F_1,25_ = 10.18, *p* < 0.05 for the effect of genotype x drug interaction) and in the hippocampus (F_1,28_ = 6.99, *p* < 0.05 for effect of genotype x fluoxetine interaction). At the same time, chronic fluoxetine treatment failed to affect *Htr1a* gene expression in the midbrain and hypothalamus of both investigated mouse lines (Figure 3A).

It was found that the *Cc2d1a* gene mRNA level was lower in the B6-M76C mice from the control group compared with those for the B6-M76B mice (F_1.27_ = 7.93, *p* < 0.001 for effect of genotype) in the midbrain. Interestingly, chronic fluoxetine treatment decreased the mRNA level of *Cc2d1a* gene in the B6-M76B mice compared with the control group (F_1.27_ = 4.47, *p* < 0.05 for effect of fluoxetine) in the midbrain. However, in the frontal cortex, hippocampus and hypothalamus, we did not find any effect of chronic fluoxetine treatment on the *Cc2d1a* gene mRNA level (Figure 3B). 

Two-way ANOVA demonstrated the effect of the genotype (F_1.20_ = 9.2, *p* < 0.01) and the drug (F_1.20_ = 9.4, *p* < 0.001) on the *Htr2a* mRNA level in the hypothalamus. Post hoc analysis showed a significant (*p* < 0.05) decrease in 5 *Htr2a* gene expression in hypothalamus of B6-M76C mice after chronic fluoxetine treatment (Figure 3C). At the same time, a fluoxetine-induced reduction in the *Htr2a* mRNA level in hypothalamus of B6-M76B mice was below the significance threshold (*p* = 0.053), whereas the mRNA baseline level in hypothalamus of B6-M76C mice was not significantly decreased compared to the B6-M76B mice (*p* = 0.062). Furthermore, the effect of the genotype (F_1.27_ = 22.1, *p* < 0.001) on the *Htr2a* mRNA level was observed in the midbrain. Chronic fluoxetine treatment led to a significant reduction (*p* < 0.05) in the *Htr2a* gene expression in the midbrain of B6-M76C mice. Additionally, B6-M76C mice from the control group demonstrated reduced (*p* < 0.05) midbrain *Htr2a* mRNA levels compared to control B6-M76B mice. Alterations of *Htr2a* gene expression in the frontal cortex caused by chronic fluoxetine treatment were at tendency levels (F_1.22_ = 3.7, *p* = 0.07 for the effect of the drug and F_1.22_ = 3.6, *p* = 0.07 for the effect of genotype x drug interaction) and post hoc analysis showed a significant (*p* < 0.05) increase in *Htr2a* gene expression in the B6-M76B mice after chronic fluoxetine treatment. No difference in *Htr2a* gene mRNA levels was found in the hippocampus of investigated animals either (Figure 3C).

B6-M76C mice demonstrated higher *Htr7* mRNA levels compared to the control B6-M76B mice (*p* < 0.05). Chronic fluoxetine treatment led to a decrease in the *Htr7* gene mRNA level in the frontal cortex of B6-M76C mice (F_1.26_ = 9.9, *p* < 0.01 for effect of genotype and drug interaction). At the same time, fluoxetine reduced the *Htr7* gene mRNA level in the hippocampus of B6-M76B mice (*p* < 0.01). The two-way ANOVA revealed the effect of genotype (F_1.27_ = 9.7, *p* < 0.01) and drug (F_1.27_ = 7.3, *p* < 0.05). Chronic fluoxetine treatment failed to affect *Htr7* mRNA levels in the midbrain and hypothalamus (Figure 3D).

Chronic fluoxetine treatment resulted in significantly (*p* < 0.01) increased expression of gene encoding key enzyme for 5-HT synthesis in the brain–tryptophan hydroxylase-2 in the midbrain of B6-M76B mice (Figure 4). Two-way ANOVA showed the effect of genotype x drug interaction (F_1.25_ = 11.1, *p* < 0.01) for the *Tph2* mRNA level in midbrain. A similar fluoxetine-induced increase was observed in B6-M76B mice for the expression of gene encoding 5-HT transporter realizing 5-HT reuptake from synaptic cleft (*Slc6a4*) (Figure 4). Two-way ANOVA revealed the effect of genotype x drug interaction (F_1.29_ = 5.5, *p* < 0.05) for the *Slc6a4* mRNA level in the midbrain. Post hoc analysis showed significant (*p* < 0.01) increase in *Slc6a4* gene expression in B6-M76B mice after chronic fluoxetine treatment. At the same time, fluoxetine treatment failed to affect the expression of the gene encoding the main enzyme for 5-HT degradation–monoamine oxidase A (*Maoa*) (Figure 4). However, B6-M76B mice from the control group demonstrated significantly higher (*p* < 0.01) expression of the *Maoa* gene compared to that in the B6-M76C mice (F_1.24_ = 19.1, *p* < 0.001 for the effect of genotype) (Figure 4).

### 2.4. Protein Levels

Chronic fluoxetine treatment led to considerable changes in 5-HT_1A_ receptor protein level. Two-way ANOVA showed the effect of genotype x drug interaction (F_1.18_ = 6.9, *p* < 0.05) for the 5-HT_1A_ receptor protein level in the hippocampus (Figure 5A). A post hoc analysis demonstrated that chronic fluoxetine administration resulted in a reduction in the 5-HT_1A_ receptor protein level in the hippocampus of B6-M76C mice (*p* < 0.01). Fluoxetine treatment failed to affect 5-HT_1A_ receptor expression in other investigated brain structures. Interestingly, the post hoc analysis also revealed increased 5-HT_1A_ receptor protein levels in the hippocampus of B6-M76C mice from the control group compared to control B6-M76B mice, although this difference was below the significance threshold (*p* = 0.087) (Figure 5A).

It is noteworthy that chronic fluoxetine treatment resulted in an increase (*p* < 0.05) in the protein level of 5-HT_1A_ receptor silencer Freud-1 in the hippocampus of B6-M76C mice but not B6-M76B mice. The two-way ANOVA showed the effect of genotype x drug interaction (F_1.15_ = 6.9, *p* < 0.05) for Freud-1 protein level in the hippocampus of B6-M76C mice (Figure 5B). Furthermore, B6-M76C mice from the control group demonstrated decreased Freud-1 expression (*p* < 0.05) in the hippocampus compared to that from control B6-M76B mice. No changes in Freud-1 expression were observed in other studied brain structures. Thus, in the hippocampus of B6-M76C mice, Freud-1 seems to play an important role in the regulation of 5-HT_1A_ receptor gene transcription. Moreover, in this brain structure Freud-1 is involved in the 5-HT_1A_ receptor gene response to chronic treatment with fluoxetine: Freud-1 protein increase was accompanied by reduction in 5-HT_1A_ receptor mRNA level.

Chronic fluoxetine administration did not affect 5-HT_2A_ and 5-HT_7_ receptors protein levels in all investigated brain structures. Control B6-M76C mice did not differ from control B6-M76B mice in the expression of these receptors (Figure 5C,D).

At the same time, the two-way ANOVA demonstrated the effect of the drug on TPH-2 protein level in the midbrain (F_1.21_ = 6.8, *p* < 0.05). Fluoxetine treatment resulted in a considerable decrease (*p* < 0.05) in the expression of this key enzyme for 5-HT biosynthesis in the brains of B6-M76B mice (Figure 6). The two-way ANOVA also showed the effect of genotype on 5-HTT protein level in the midbrain (F_1.16_ = 4.8, *p* < 0.05). However, the post hoc analysis retrieved no significant changes in 5-HTT expression. Chronic fluoxetine treatment failed to affect the MAO A protein level in the midbrain of both investigated mouse lines as well (Figure 6). 

### 2.5. The Levels of 5-HT, 5-HIAA and 5-HIAA/5-HT Ratio

Chronic fluoxetine administration led to a reduction in the 5-HT level in the frontal cortex and midbrain of both mouse lines compared with control groups (F_1.24_ = 28.65, *p* < 0.001 for effect of the drug in frontal cortex and F_1,23_ = 36.52, *p* < 0.001 for midbrain). At the same time, fluoxetine treatment resulted in a decrease in 5-HT level in the hippocampus only in B6-M76B mice (F_1.24_ = 14.50, *p* < 0.001 for effect of fluoxetine). In hypothalamus, fluoxetine did not affect 5-HT levels in both lines; however, the level of neurotransmitters was higher in B6-M76C mice from the control group compared to that for the B6-M76B control animals (F_1.23_ = 12.66, *p* < 0.01 for effect of genotype) (Figure 7A).

Fluoxetine administration reduced 5-HIAA in frontal cortex of both lines compared with the control groups (F_1.24_ = 17.96 *p* < 0.001 for effect of fluoxetine). At the same time, chronic antidepressant treatment decreased 5-HIAA levels in the hippocampus and midbrain only in B6-M76B mice (F_1.24_ = 5.33, *p* < 0.05 for effect of fluoxetine in hippocampus and F_1.23_ = 18.40, *p* < 0.001 for midbrain). In the hypothalamus, fluoxetine failed to affect 5-HIAA levels in both lines (Figure 7B).

The 5-HIAA/5-HT ratio reflecting 5-HT turnover was not altered followed chronic fluoxetine treatment in all investigated structures of both mouse lines (Figure 7C).

## 3. Discussion

Recombinant B6-M76C mice differed from control B6-M76B mice only by CBA-derived 102.73–110.56 Mbp fragment of chromosome 13 on identical genetic backgrounds demonstrated paradoxical pro-depressive responses to chronic treatment with the classical SSRI fluoxetine (Table 1). The obtained results show that small changes in genetic background in B6-M76C mice lead to dramatic alteration of the behavioral response to the classic antidepressant fluoxetine. Fluoxetine treatment failed to produce an antidepressive effect in B6-M76B mice. Taking into account that B6-M76B mice carry a C57BL/6 genetic background, our data are in general agreement with other studies that have reported marked strain and genotype dependence in response to SSRIs in a forced swim test [32,33,34,35,36]. 

Recently, we found that the changes in genetic background of B6-M76C mice altered the sensitivity of 5-HT_1A_ receptors to chronic activation with 5-HT_1A_ receptors agonist 8-OH-DPAT, which allowed us to suggest the altered sensitivity of presynaptic 5-HT_1A_ receptors in these animals [31]. The results of the current study allow us to suggest that fluoxetine led to the activation rather than desensitization of presynaptic 5-HT_1A_ receptors in B6-M76C mice after two weeks of exposure. This inhibits the brain’s 5-HT system functional activity and, hence, produces a pro-depressive effect. Similar pro-depressive effects are sometimes observed in humans, especially in the first few weeks after starting treatment [37]. 

Although fluoxetine decreased the brain 5-HT and 5-HIAA levels (Table 1), it failed to affect the 5-HIAA/5-HT ratio, reflecting 5-HT turnover in both investigated mouse lines. One could suggest that the decrease in total 5-HT and 5-HIAA levels reflects a “normal” response to chronic SSRI treatment in B6-M76B mice. In this regard, the unchanged 5-HT and 5-HIAA levels in the hippocampus of B6-M76C mice could reflect the incorrect response of B6-M76C mice to chronic fluoxetine treatment.

Our data on the absence of fluoxetine effect on 5-HT turnover are mainly in line with the literature, showing an initial increase in 5-HT levels and a return to baseline or even lower levels after prolonged SSRI treatment [38,39,40]. In mice expressing the mutation in *Tph2* gene (R439H Tph2 KI), chronic treatment with fluoxetine resulted in a dramatic depletion of 5-HT while having little effect on the wild-type control [41]. However, in albino mice, the 5-HIAA/5-HT ratio was significantly elevated after 2–3 weeks of fluoxetine treatment [42]. The mentioned results indicate that the changes in levels of 5-HT, 5-HIAA and 5-HIAA/5-HT in the brains of mice in response to fluoxetine treatment can vary greatly depending on their genetic background.

We revealed significant differences in response of 5-HT elements to chronic fluoxetine treatment between investigated mouse lines (Table 1). Chronic fluoxetine treatment altered the expression of the key regulator [4] of brain 5-HT system functional activity—5-HT_1A_ receptor. Fluoxetine produced a decrease in hippocampal 5-HT_1A_ receptor gene expression in B6-M76C mice, accompanied by a reduction in 5-HT_1A_ receptor protein level in this brain structure. In the frontal cortex, fluoxetine induced a decrease in *Htr1a* mRNA levels that was not coupled with corresponding changes in receptor protein levels. It is interesting to note that fluoxetine affected 5-HT_1A_ receptors only in B6-M76C mice and not in B6-M76B mice. Furthermore, it is noteworthy that chronic SSRI application affected hippocampal and cortical but not midbrain 5-HT_1A_ receptors. Given a predominant localization of presynaptic 5-HT_1A_ receptors in midbrain raphe nuclei area and postsynaptic 5-HT_1A_ receptors in other brain structures, especially the hippocampus, where the highest expression of 5-HT_1A_ receptors is observed [43], these data indicate the reduced sensitivity of presynaptic and increased sensitivity of postsynaptic 5-HT_1A_ receptors in B6-M76C mice. Additionally, the higher *Htr1a* mRNA levels in the frontal cortex and hippocampus of B6-M76C mice compared to B6-M76B mice indirectly confirms the suggestion of increased sensitivity of postsynaptic 5-HT_1A_ receptors in B6-M76C mice as well. 

Importantly, the chronic fluoxetine treatment-induced decrease in hippocampal 5-HT_1A_ receptor expression in B6-M76C mice indicates a reduction in 5-HT neurotransmission. This 5-HT_1A_ receptor decrease may explain the pro-depressive effect induced by chronic application of fluoxetine. 

Chronic fluoxetine treatment resulted in reduction in *Cc2d1a* gene mRNA level in B6-M76B mice in the midbrain. However, these changes were not accompanied by alterations in Freud-1 protein levels. At the same time, fluoxetine increased the protein level for 5-HT_1A_ receptor silencer Freud-1 in the hippocampus of B6-M76C mice. This result, taken together with fluoxetine-induced reduction in the *Htr1a* mRNA level in the hippocampus of B6-M76C mice as well as with the data on decreased Freud-1 expression in the hippocampus of B6-M76C mice from control group, is in agreement with the important role of Freud-1 in the regulation of 5-HT_1A_ receptor gene expression [22]. These results indicate that Freud-1-mediated transcriptional regulation of hippocampal 5-HT_1A_ receptor gene seems to play an important role in the response of B6-M76C mice to chronic fluoxetine treatment.

It is well known that the functional state of 5-HT_1A_ receptors is under the control of other serotonin receptors—for example, 5-HT_2A_ and 5-HT_7_ [16,25,26,27,28]. Chronic fluoxetine treatment produced significant changes in *Htr2a* and *Htr7* mRNA levels. However, these alterations were not accompanied by changes in 5-HT_2A_ and 5-HT_7_ receptor protein levels, which hampers the understanding of these receptors’ role in the 5-HT_1A_ receptor response to chronic SSRI application.

Fluoxetine considerably affects expression of the key enzyme for 5-HT biosynthesis in the brain TPH-2 in B6-M76B mice only. Interestingly, fluoxetine treatment resulted in an increase in *Tph2* gene mRNA level accompanied by a reduction in TPH-2 protein level in the midbrain of B6-M76B mice (Table 1). The latter seems to reflect compensatory mechanisms directed to diminish the fluoxetine-induced effect of 5-HTT blockade on the 5-HT system. Likely, similarly directed compensatory mechanisms explain the increase in *Slc6a4/*5-HTT gene expression in B6-M76B mice followed by chronic 5-HTT blockade. However, these changes were not coupled to 5-HTT protein level alterations. Nevertheless, in B6-M76C mice with altered sensitivity of 5-HT_1A_ receptors, chronic fluoxetine treatment failed to affect key enzymes for 5-HT synthesis and catabolism as well as 5-HT transporter, which also indicate a reduced response to chronic fluoxetine treatment.

Thus, individual differences in genetic background may result in reduced sensitivity or even an inversed response to a classic antidepressant from the SSRI family fluoxetine, which is known to be widely utilized in clinics. Our data draw attention to the 5-HT_1A_ autoreceptor as a hotspot in the mechanisms of antidepressant resistance. Moreover, B6-M76C mice demonstrating a pro-depressive response to chronic treatment with fluoxetine seem to represent a model for investigation of mechanisms underlying antidepressant resistance. 

## 4. Materials and Methods

### 4.1. Animals

Adult (10–12 weeks old) male mice of B6-M76C and B6-M76B lines [31] were used. The animals (about 25 g) were housed under standard conditions (20–22 °C, food and water ad libitum, 12 h light/dark cycle) in groups of 7–8 per cage (40 cm × 25 cm × 15 cm). The mice were isolated into individual cages two days before behavioral tests to remove the group effect. The breeding of B6-M76C and B6-M76B lines was conducted in the Center for Genetic Resources of Laboratory Animals at Institute of Cytology and Genetics, Siberian Division, Russian Academy of Sciences (ICG SB RAS), supported by the Ministry of Education and Science of Russia (unique identifier of the project, RFMEFI62117 × 0015). 

All experimental procedures were in compliance with the Guide for the Care and Use of Laboratory Animals, Eighth Edition, Committee for the Update of the Guide for the Care and Use of Laboratory Animals; National Research Council © 2020 National Academy of Sciences (USA) for animal experiments and the trial was approved by the ICG SB RAS ethics committee and registered in ICG SB RAS (Protocol No. 34, 15.06.2016). 

### 4.2. Drug

The classical Selective Serotonin Reuptake Inhibitor fluoxetine (Biokom, Stavropol, Russia) was dissolved in saline and administered intraperitoneally (i.p.) in a dose of 20 mg/kg.

### 4.3. Design of the Experiment

The effects of chronic fluoxetine treatment were assessed after 14 days of daily i.p injections. We used thirty B6-M76C mice and thirty-two B6-M76B mice. Mice of each line were divided into two equal groups (*n* = 15–17 per group)—a control group and an experimental group (fluoxetine). Control mice were treated with the same volume of saline. Their behavior was ad libitum tested through open field (OF) and forced swim tests (FST) the next day after the last fluoxetine dose to avoid acute administration effects.

### 4.4. Open Field Test

The OF test was carried out in a circle arena (40 cm in diameter) surrounded by a white plastic wall (25 cm high) and illuminated through the mat and semi-transparent floor with two halogen lamps of 12 W each placed 40 cm under the floor [44]. The mouse was placed near the wall and its movements were tracked for 5 min with a digital camera (Sony, Tokyo, Japan). The area was carefully cleaned after each test. The video stream from the camera was analyzed frame by frame using the original EthoStudio software [45]. The horizontal locomotor activity (distance run) and time in the center were measured automatically. 

### 4.5. Forced Swim Test

Mice were placed in a clear glass box (30 × 30 × 30 cm) filled with water at a temperature of 25 °C. Mouse mobility was measured during 4 min (after 2 min adaptation) by the EthoStudio program. The program measured the rate of change in the silhouette of an animal, which was determined as the number of animal-associated pixels changed between two adjacent frames [46]. For the behavioral tests (OF test and FST), 62 animals were used.

### 4.6. Brain Structures Isolation

Two days after behavioral testing, animals were decapitated and the brain structures (frontal cortex, hippocampus, midbrain and hypothalamus) were isolated on ice, frozen in liquid nitrogen and stored at −80 °C until following procedures. 

### 4.7. Tissue Extraction for Isolation Total mRNA and for HPLC

The brain structures were homogenized in the Potter-Elvehjem homogenizer in 200 µL 50 mM Tris HCl, pH 7.6 (4 °C). Then, 150 μL aliquot was used for total RNA extraction with Trizol and 50 μL aliquot was mixed with 0.6 M HClO4 for HPLC (see below). For the experiment, 28 animals were used.

### 4.8. Real-Time PCR

Total RNA was isolated with Trizol reagent (Thermoscientific, Waltham, MA, USA) and 1 µg of the mRNA was used for cDNA synthesis with a random hexanucleotide primer. The number of copies of *Htr1a, Htr2a, Htr7, Slc6a4* (gene coding 5-HTT), *Tph2, Maoa* and *Cc2d1a* genes’ cDNA was estimated using SYBR Green (Synergy Bands, Inc.^®^, New York, NY, USA) real-time quantitative PCR with selective primers (Table 2). We used 50, 100, 200, 400, 800, 1600, 3200 and 6400 copies of genomic DNA as external standards for all studied genes. The mRNA levels were presented as the number of its cDNA copies with respect to 100 copies of *Polr2a* cDNA [47,48,49]. 

### 4.9. HPLC Protocol

For HPLC, 50 μL aliquot was mixed with 0.6 M HClO_4_ (Sigma–Aldrich, St. Louis, MO, USA) containing 200 ng/mL isoproterenol (Sigma Aldrich, St. Louis, MO, USA) as an internal standard. Homogenate was centrifuged at 12,000× *g* for 15 min at 4 °C for protein precipitation. The supernatants were diluted two times with ultrapure water and filtered using a centrifuge tube with 0.22 µm cellulose acetate filter (Spin-X^®^, Mooresville, NC, USA). The pellet was stored at −20 °C until protein quantitation by the Bradford method. Twenty microliters of the filtered supernatant were injected into the loop of the HPLC system.

The levels of 5-HT and 5-HIAA were analyzed in the brain structures using HPLC as it was described earlier [50]. 

The temperature of the column was stabilized at 40 °C. The amounts (ng) of substances were calculated relative to the internal standard. The contents of substances were expressed in ng/mg of protein (assayed by Bradford). 

### 4.10. Western blot

The protein levels were estimated as described earlier [50]. The used primary antibodies are presented in Table 3. Quantification of protein bands was performed by ImageStudio (LI-COR, Lincoln, NE, USA). Target protein levels were normalized to GAPDH chemiluminescence relative units, represented as the percentage of sham B6-M76B animals. For the experiment, 27 animals were used.

### 4.11. Statistical Analysis

All values were presented as means ± SEM and compared with a two-way factorial ANOVA with genotype (B6-M76B vs. B6-M76C) and fluoxetine (sham vs. fluoxetine) as between factors followed by Fisher’s post hoc analysis. Outliers were determined using the Dickson parameter and excluded from the analysis [51]. Statistical significance was set at *p* < 0.05.

## Figures and Tables

**Figure 1 ijms-21-08784-f001:**
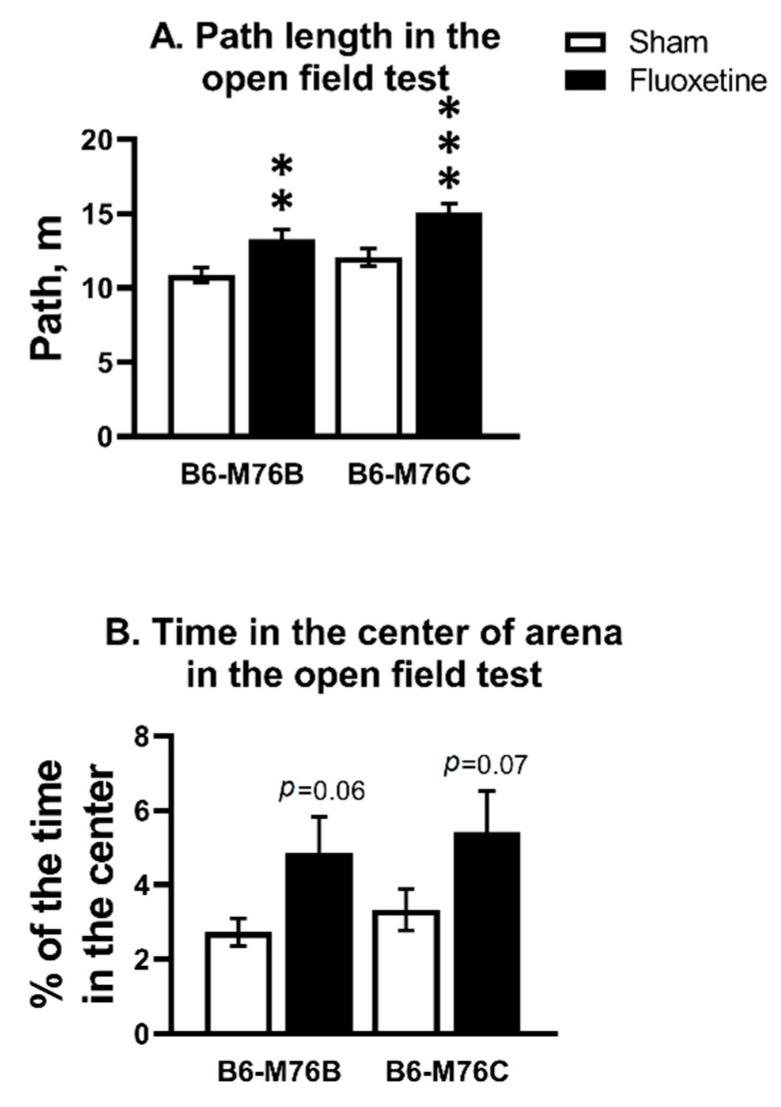
Effect of chronic fluoxetine administration on path length (**A**) and time in the center of arena (**B**) in the open field test. *n* ≥ 15 for each group. All values are presented as mean ± SEM. ** *p* < 0.01, *** *p* < 0.001 compared to control mice of the same line.

**Figure 2 ijms-21-08784-f002:**
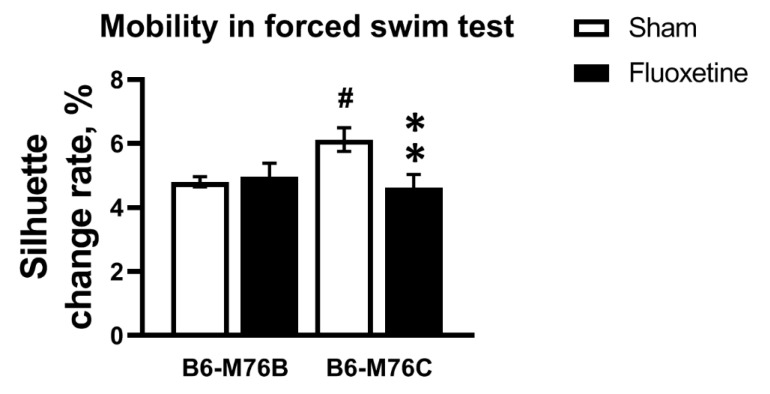
Effect of chronic fluoxetine administration on mobility (estimated as silhouette change rate) in a forced swim test. *n* ≥ 15 for each group. All values are presented as mean ± SEM. ** *p* < 0.01 compared to control mice of the same line, # *p* < 0.05 compared to control B6-M76B mice.

**Figure 3 ijms-21-08784-f003:**
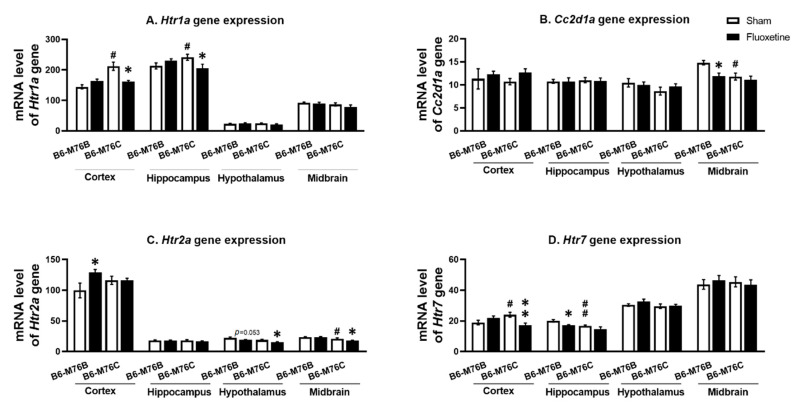
*Htr1a* (**A**), *Htr2a* (**C**) and *Htr7* (**D**) receptor and *Cc2d1a* (**B**) genes expression in the brain structures of control and chronically-treated-with-fluoxetine B6-M76C and B6-M76B mice. Gene expression is presented as the number of complementary DNA (cDNA) copies with respect to 100 cDNA copies of *rPol2a*, *n* ≥ 8 for each group. All values are presented as mean ± SEM. * *p* < 0.05, ** *p* < 0.01 compared to control mice of the same line, # *p* < 0.05, ## *p* < 0.01 compared to control B6-M76B mice.

**Figure 4 ijms-21-08784-f004:**
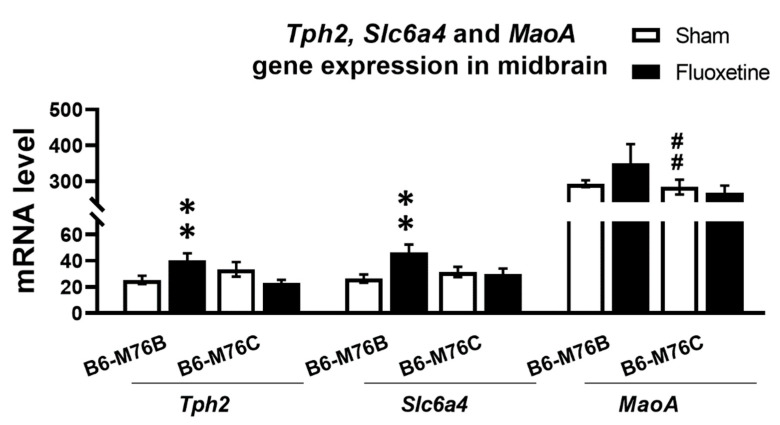
*Tph2*, *Slc6a4* and *Maoa* genes expression in the midbrain of control and chronically-treated-with-fluoxetine B6-M76C and B6-M76B mice. Gene expression is presented as the number of cDNA copies with respect to 100 cDNA copies of *rPol2a*, *n* ≥ 8 for each group. All values are presented as mean ± SEM. ** *p* < 0.01 compared to control mice of the same line, ## *p* < 0.01 compared to control B6-M76B mice.

**Figure 5 ijms-21-08784-f005:**
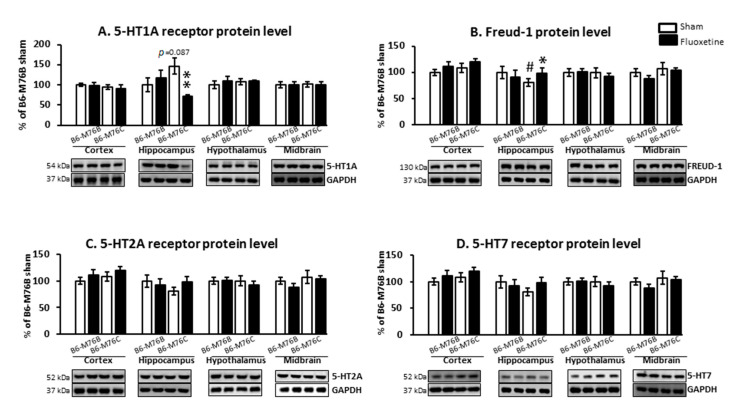
Protein 5-HT_1A_ (**A**), 5-HT_2A_ (**C**), 5-HT_7_ (**D**) and Freud-1 (**B**) levels in the brain structures of control and chronically-treated-with-fluoxetine B6-M76C and B6-M76B mice. Protein levels were assessed in chemiluminescence relative units and normalized to Glyceraldehyde 3-phosphate dehydrogenase (GAPDH) chemiluminescence relative units. *n* = 7 for each group. All values are presented as mean ± SEM. * *p* < 0.05, ** *p* < 0.01 compared to control mice of the same line, # *p* < 0.05 compared to control B6-M76B mice.

**Figure 6 ijms-21-08784-f006:**
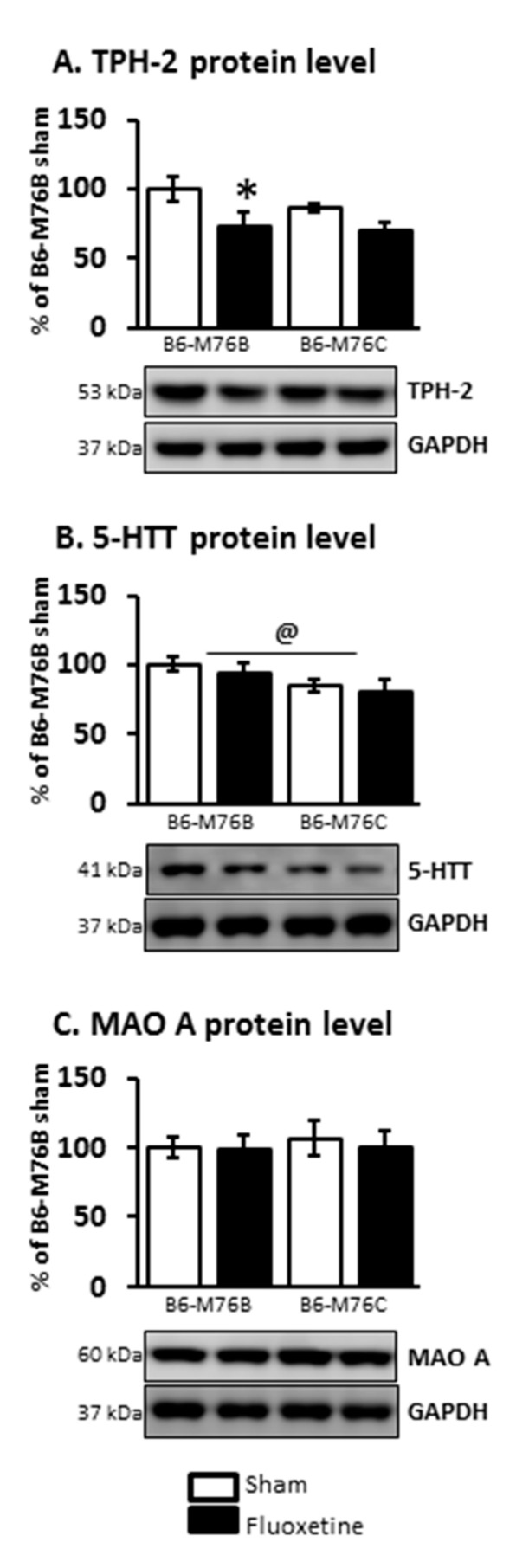
TPH-2 (**A**), 5-HTT (**B**) and MAO A (**C**) protein levels in the brain structures of control and chronically-treated-with-fluoxetine B6-M76C and B6-M76B mice. Protein levels were assessed in chemiluminescence relative units and normalized to GAPDH chemiluminescence relative units. *n* = 7 for each group. All values are presented as mean ± SEM. * *p* < 0.05 compared to control mice of the same line, @ *p* < 0.05 as effect of genotype.

**Figure 7 ijms-21-08784-f007:**
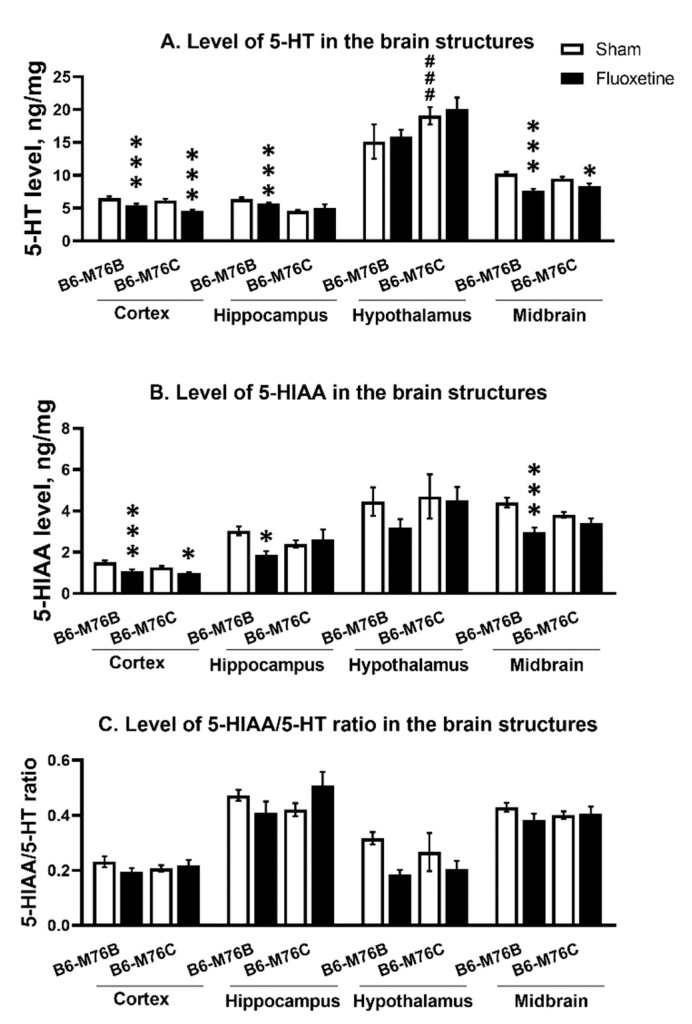
Level of 5-HT (**A**) 5-HIAA levels (**B**) and 5HIAA/5-HT ratio (**C**) in the brain structures of control and chronically-treated-with-fluoxetine B6-M76C and B6-M76B mice. Levels of 5-HT and 5-HIAA are presented in ng/mg, *n* ≥ 8 for each group. All values are presented as mean ± SEM. *** *p* < 0.001; * *p* < 0.05 compared to control mice of the same line; ### *p* < 0.001 compared to control B6-M76B.

**Table 1 ijms-21-08784-t001:** Summary of the obtained results.

		B6-M76B CFT Group Compared to Sham	B6-M76C CFT Group Compared to Sham
Open field test	Path length	↑	↑
	Time in center	=	=
Forced swim test	Mobility	=	↓
5-HT_1A_ mRNA/Protein	Frontal cortex	=/=	↓/=
	Hippocampus	=/=	↓/↓
Freud-1 mRNA/Protein	Hippocampus	=/=	=/↑
	Midbrain	↓/=	=/=
5-HT_2A_ mRNA/Protein	Frontal cortex	↑/=	=/=
	Hypothalamus	↓/=	↓/=
	Midbrain	=/=	↓/=
5-HT_7_ mRNA/Protein	Frontal cortex	=/=	↓/=
	Hippocampus	↓/=	=/=
TPH2 mRNA/Protein	Midbrain	↑/↓	=/=
5-HTT mRNA/Protein	Midbrain	↑/=	=/=
MAO A mRNA/Protein	Midbrain	=/=	=/=
5-HTT level	Frontal cortex	↓	↓
	Hippocampus	↓	=
	Hypothalamus	=	=
	Midbrain	↓	↓
5-HIAA level	Frontal cortex	↓	↓
	Hippocampus	↓	=
	Hypothalamus	=	=
	Midbrain	↓	=

↓—decrease, ↑—increase, = not changed.

**Table 2 ijms-21-08784-t002:** The primer sequences, annealing temperatures and PCR product lengths.

Gene	Sequence	Annealing Temperature, °C	Product Length, bp
*Htr1a*	F 5′-ctgtgacctgtttatcgccctg-3′ R 5′-gtagtctatagggtcggtgattgc-3′	62	200
*Htr2a*	F 5′-agaagccaccttgtgtgtga-3′ R 5′-ttgctcattgctgatggact-3′	61	169
*Htr7*	F5′-ggctacacgatctactccaccg-3′ R5′-cgcacactcttccacctccttc-3′	65	198
*Tph2*	F 5′-cattcctcgcacaattccagtcg-3′ R 5′- cttgacatattcaactagacgctc -3′	61	239
*Slc6a4*	F 5′-cgctctactacctcatctcctcc-3′ R 5′- gtcctgggcgaagtagttgg -3′	63	101
*Maoa*	F 5′-aatgaggatgttaaatgggtagatgttggt-3′ R 5′-cttgacatattcaactagacgctc-3′	61	138
*Cc2d1a*	F 5′-gcaaagccgggcaacatcatc-3′ R 5′-tagcagaggtgggtgtagtgg-3′	60	181
*rPol2*	F 5′-tgtgacaactccatacaatgc-3′ R 5′-ctctcttagtgaatttgcgtact-3′	60	194

**Table 3 ijms-21-08784-t003:** List of antibodies used and immunodetection conditions.

Antibodies, Manufacturer	Breeding	Incubation Time, Conditions
Rabbit polyclonal primary antibodies to 5-HT_1A_ protein, Abcam, United Kingdom, ab85615	1:1000 in 5% milk powder with Tris-Buffered Saline with Tween 20 (TBST)	Night at 4 °C
Rabbit polyclonal primary antibodies to 5-HT_2A_ protein, Novus Biologicals, USA, Novus NBP1-49172	1:250 in 5% milk powder with TBST	Night at 4 °C
Rabbit monoclonal primary antibodies to 5-HT_7_ protein, Abcam, United Kingdom, ab128892	1:500 in 5% milk powder with TBST	Night at 4 °C
Rabbit monoclonal primary antibodies to Freud-1 protein, Abcam, United Kingdom, ab191472	1:2000 in 5% milk powder with TBST	Night at 4 °C
Rabbit polyclonal primary antibodies to TPH-2 protein, Abcam, United Kingdom, ab111828	1:1000 in 5% milk powder with TBST	Night at 4 °C
Rabbit polyclonal primary antibodies to 5-HTT protein, US Biological Life Sciences, 303614	1:1000 in 5% milk powder with TBST	Night at 4 °C
Rabbit monoclonal primary antibodies to MAOA protein, Abcam, United Kingdom, ab126751	1:250 in 5% milk powder with TBST	Night at 4 °C
Rabbit polyclonal primary antibodies to GAPDH protein, conjugated to horseradish peroxidase, Santa Cruz, USA, sc-25778	1:500 in 5% milk powder with TBST	2 h at Room Temperature
Secondary goat antibodies against rabbit immunoglobulins conjugated to horseradish peroxidase, Invitrogen, USA, G-21234	1:10,000 in 5% milk powder with TBST	1 h at Room Temperature

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
