# Peer review of "Genetic Background Underlying 5-HT1A Receptor Functioning Affects the Response to Fluoxetine"

_ijms, 2020, doi:10.3390/ijms21228784_

Round 1

Reviewer 1 Report

In this manuscript authors studied the effect of antidepressant fluoxetine treatment in two different mouse models. Authors also suggest that genetic background can affect the effect of treatment and some of the results are interesting. Here are the comments, which authors need to address in the revised manuscript.

  1. Authors should provide a better explanation why fluoxetine failed to affect 5-HIAA/5-HT ratio in both investigated mouse lines.
  2. Authors showed that percentage of time spent in the center of the arena was not changed in both lines (Figure. 1B). However, the differences are clearly noticeable. What are the p values for these analysis (Fig. 1B).

Author Response

Thank you for your revision.

We agree with all your comments. We corrected text of article according with your remarks.

Comments and Suggestions for Authors

  1. Authors should provide a better explanation why fluoxetine failed to affect 5-HIAA/5-HT ratio in both investigated mouse lines.

Answer: Fluoxetine decreased both 5-HT and 5-HIAA levels. The similarity of the reduction of 5-HT and 5-HIAA is the reason on unchanged 5-HIAA/5-HT ratio indicating undisturbed synchronization of the 5-HT synthesis/degradation processes.

Our data on the absence of fluoxetine effect on 5-HT turnover is mainly in the line with literature showing initial increase in 5-HT levels and return to baseline or even lower levels after prolonged SSRI treatment (Caccia et al., 1993, Bianchi et al., 2002, Popa et al., 2010). In mice expressing the mutation in Tph2 gene (R439H Tph2 KI) chronic treatment fluoxetine resulted in a dramatic depletion of 5-HT level while having little effects in wild-type control (Siesser et al., 2013). But in albino mice the 5-HIAA/5-HT ratio was significantly elevated after 2–3 weeks of fluoxetine treatment (Stenfors and Ross, 2002). Mentioned results indicate that the changes in levels of 5-HT, 5-HIAA and 5-HIAA/5-HT in the brain of mice in response to fluoxetine treatment can vary greatly depending on the genetic background.

This paragraph was added to the MS.

  1. Authors showed that percentage of time spent in the center of the arena was not changed in both lines (Figure. 1B). However, the differences are clearly noticeable. What are the p values for these analysis (Fig. 1B).

Answer: We added to the results the statistical analysis of the percentage of time spent in the center and changed the Figure 1B.

In the MS we added: “Statistical analysis revealed effect of fluoxetine; F1.58 =7.04, p<0.05 for percentage of time spent in the center of the arena but post –hoc analysis shown only tendency for both B6-M76C and B6-M76B mice compared with correspondent control groups (Figure. 1B)”.

Reviewer 2 Report

In this report, the authors investigated the effect of chronic antidepressant (fluoxetine) treatment in recombinant B6-M76C mice, which have disrupted functioning of 5-HT1A autoreceptors. They found fluoxetine produced pro-depressive effect and changes in brain serotonin-related mRNA/protein levels in B6-M76C mice compared with the controls (B6-M76B mice). The paper is of interest because it showed that changes in 5-HT1A genetic background may have a dramatic effect on antidepressant treatment response, however a number of issues should be considered.

  1. Fig 2A and 2B have the same title that I cannot tell the difference between them. It seems that fluoxetine treatment has no antidepressive effect (increased mobility) in control group (B6-M76B mice). That’s quite different from previous studies. Please discuss that. How did the authors decide the fluoxetine dose (20 mg/kg.)?
  2. P 195. “Chronic fluoxetine treatment led to decrease of mobility only in B6-M76C mice.” It looks that fluoxetine treatment led to normalize the increased mobility in B6-M76C mice rather than resulting in depressive-like behavior. Fluoxetine-treated B6-M76C mice showed similar mobility as the two B6-M76B groups.
  3. P209. “Notably, intact B6-M76C mice demonstrated higher level of Htr1a gene mRNA level compared with B6-M76B mice (p<0.05).” in whole brain or in specific brain region?
  4. The findings are very complicated. It would be better to show the findings in a Table.
  5. Since the 5-HT1A genetic background may have some effect on antidepressant treatment response, it’s better to cite some related clinical pharmacogenetic studies.
  6. Abbreviations should be defined on their first appearance in the text (e.g. SSRI, OF, CFT).

Author Response

Thank you for your revision

We agree with all your comments. We corrected text of article according with your remarks Comments and Suggestions for Authors

  1.  Fig 2A and 2B have the same title that I cannot tell the difference between them.

Answer: We apologize for incorrect Figure 2. Figure 2A was removed.

  1. It seems that fluoxetine treatment has no antidepressive effect (increased mobility) in control group (B6-M76B mice). That’s quite different from previous studies. Please discuss that.

Answer: Indeed, in B6-M76B mice fluoxetine had no effect on mobility. This data obtained for the first time on this mouse model. In literature one could find that some SSRIs have genotype-depended effect on behavior in the FST (Lucki et al., 2001; David et al., 2003; Cervo et al., 2005; Guzzetti et al., 2008; Kulikov at al., 2011). Several studies have reported marked strain differences in the response to SSRIs in the FST (Lucki et al., 2001; David et al., 2003). Lucki et al., 2001 have shown that fluoxetine failed to produce significant effects on behavior of the C57BL/6J mice. Since B6-M76B mice carry genetic background of C57BL/6J mice, our data in agreement with the results obtained by Lucki et al 2001. We added this data to the discussion.

  1. Please discuss that. How did the authors decide the fluoxetine dose (20 mg/kg.)?

Answer: We have chosen the dose 20 mg/kg according to literature data (Lucki et al., 2001, Tikhonova et al 2009, Tikhonova et al 2010) since this dose utilized in the number of studies (PMID 32768191; 32754030; 32218734; 31819774).

  1. P 195. “Chronic fluoxetine treatment led to decrease of mobility only in B6-M76C mice.” It looks that fluoxetine treatment led to normalize the increased mobility in B6-M76C mice rather than resulting in depressive-like behavior. Fluoxetine-treated B6-M76C mice showed similar mobility as the two B6-M76B groups.

Answer: We agree that mobility of two B6-M76B groups did not differ from fluoxetine-treated B6-M76C mice. At first glance the results could be interpreted as a normalization of the increased mobility in B6-M76C after chronic fluoxetine treatment. However, B6-M76B and B6-M76C mice did not differ by locomotor activity and fluoxetine even increased locomotion in both investigated mouse lines (Fig 1). The latter indicate that reduction of mobility in FST is related more to depressive-like behavior rather than to locomotor activity. Anyway, fluoxetine-induced changes in the mobility in FST is inverted that indicate that small changes in the genetic background could lead to inversion of the response to chronic treatment with SSRIs.

  1. P209. “Notably, intact B6-M76C mice demonstrated higher level of Htr1a gene mRNA level compared with B6-M76B mice (p<0.05).” in whole brain or in specific brain region?

Answer: Differences in Htr1a gene expression were observed in cortex and hippocampus. We removed “Notably, intact B6-M76C mice demonstrated higher level of Htr1a gene mRNA level compared with B6-M76B mice (p<0.05)” phrase from the text.

  1. The findings are very complicated. It would be better to show the findings in a Table.

Answer: We added table summarizing the data in the discussion.

  1. Since the 5-HT1A genetic background may have some effect on antidepressant treatment response, it’s better to cite some related clinical pharmacogenetic studies

Answer: We added to introduction some data from preclinical and clinical pharmacogenetic studies (Stockmeier et al., 1998, Boldrini et al., 2008; Zhang et al., 2010; Blier and Ward, 2003; Gunther et al., 2011)

  1. Abbreviations should be defined on their first appearance in the text (e.g. SSRI, OF, CFT)

Answer: Abbreviation CFT was removed from the MS. Other abbreviations were added on their first appearance in the text.

Round 2

Reviewer 2 Report

The authors responded adequately to the my comments.